# Physiological and Molecular Responses of Woody Plants Exposed to Future Atmospheric CO_2_ Levels under Abiotic Stresses

**DOI:** 10.3390/plants11141880

**Published:** 2022-07-20

**Authors:** Ana Karla M. Lobo, Ingrid C. A. Catarino, Emerson A. Silva, Danilo C. Centeno, Douglas S. Domingues

**Affiliations:** 1Department of Biodiversity, Institute of Biosciences, São Paulo State University, UNESP, Rio Claro 13506-900, Brazil; ingrid.catarino@unesp.br; 2Institute of Environmental Research, São Paulo 04301-002, Brazil; easilva@sp.gov.br; 3Centre for Natural and Human Sciences, Federal University of ABC, São Bernardo do Campo 09606-045, Brazil; danilo.centeno@ufabc.edu.br

**Keywords:** climate change, multiple stresses, trees

## Abstract

Climate change is mainly driven by the accumulation of carbon dioxide (CO_2_) in the atmosphere in the last century. Plant growth is constantly challenged by environmental fluctuations including heat waves, severe drought and salinity, along with ozone accumulation in the atmosphere. Food security is at risk in an increasing world population, and it is necessary to face the current and the expected effects of global warming. The effects of the predicted environment scenario of elevated CO_2_ concentration (e[CO_2_]) and more severe abiotic stresses have been scarcely investigated in woody plants, and an integrated view involving physiological, biochemical and molecular data is missing. This review highlights the effects of elevated CO_2_ in the metabolism of woody plants and the main findings of its interaction with abiotic stresses, including a molecular point of view, aiming to improve the understanding of how woody plants will face the predicted environmental conditions. Overall, e[CO_2_] stimulates photosynthesis and growth and attenuates mild to moderate abiotic stress in woody plants if root growth and nutrients are not limited. Moreover, e[CO_2_] does not induce acclimation in most tree species. Some high-throughput analyses involving omics techniques were conducted to better understand how these processes are regulated. Finally, knowledge gaps in the understanding of how the predicted climate condition will affect woody plant metabolism were identified, with the aim of improving the growth and production of this plant species.

## 1. Introduction

The concentration of greenhouse gases in the atmosphere has changed in the last millennium, mostly as a consequence of anthropogenic activities since the Industrial Revolution. The global warming effect observed from the middle 20th century is mainly driven by the high accumulation in the concentration of carbon dioxide (CO_2_) and other gases, such as methane (CH_4_), nitrous oxide (N_2_O) and ozone (O_3_), in the atmosphere [1]. Fossil-fuel burning and land-use change have contributed to the atmospheric CO_2_ rise over the years, from 283 ppm in the 1800s to the current global average level of ~418 ppm in 2022 [2,3,4]. As the emission rate of CO_2_ continues around the world, an increase of ~140 ppm is expected in the mid-century, which will probably reach ~1000 ppm by 2100 [5]. Additionally, climate changes involving higher temperatures (up to 4.8 °C) and extreme environmental conditions, such as severe drought, flooding, cold and heatwave events, are predicted to negatively impact all living species on the planet [6,7,8,9].

In this scenario, plant growth, phenological phases and development will be frequently challenged by climatic fluctuations, leading food security and food quality to be at risk in an increasing world population [10,11,12]. A range of studies over the years has been developed in an attempt to understand how plants will cope with elevated CO_2_ (e[CO_2_]), isolated or combined with other abiotic stresses, to identify new approaches to increase global sustainability and crop yield [13,14,15,16,17,18]. However, most of these are focused on photosynthesis and the growth rate of annual crops, leaving a gap in knowledge involving woody plants, especially in integrating physiological and molecular data. Besides their role in the ecosystem, many tree species have particular importance for worldwide agriculture production, and little is known about the effects of e[CO_2_] and other environmental stresses on their metabolism [19,20].

In the literature, the experiments involving woody plants exposed to e[CO_2_] are fragmented and controversial. The main reasons for these controversial results could be related to the short-term experiments considering the perennial characteristics of woody plants, the limitations to carrying out long-term experiments (e.g., pots and chambers size) and the high costs of field experiments. Nevertheless, in general, the responses include physiological, biochemical, molecular and morphological modifications which, in many cases, increase photosynthesis and water use efficiency and trigger stress defense mechanisms such as ROS (reactive oxygen species) scavenging [21,22,23,24,25,26]. Whether these responses activated by e[CO_2_] make woody plants more resistant to abiotic stresses and how the whole plant metabolism is regulated under these conditions are under investigation. We hypothesized that finding markers induced by e[CO_2_] is promising to improve the growth of woody plants exposed to adverse environmental conditions. Therefore, we outline here an integrative view of the effects of e[CO_2_] and other abiotic stresses on the growth and development of woody plants, specially selecting studies using large-scale “omics” technologies to depict molecular responses to e[CO_2_] to better understand how woody plants will cope with climate change (Appendix A).

## 2. Woody Plant Growth and Development under Elevated CO_2_

### 2.1. Effect on Leaf Photosynthesis

Atmospheric e[CO_2_] usually stimulates the source activity of woody plants, but the response varies with species and exposure time. Initially, photosynthesis can be enhanced by the higher CO_2_ availability surrounding ribulose-1,5-bisphosphate (RuBP) carboxylase/oxygenase (Rubisco). Under these conditions, the carboxylation role of Rubisco is favored to the detriment of its oxygenation activity, increasing photosynthetic efficiency and the production of sugars. However, in some species, after an acclimation period, these reactions slow down or decrease when compared to the initial levels, mostly as a consequence of stomatal closure and/or soluble sugar accumulation in the mesophyll cells [27,28]. According to the FvCB model, photosynthesis can be limited biochemically by Rubisco activity, RuBP regeneration or triose phosphate utilization (TPU) [29,30,31]. Hence, as Rubisco’s substrate CO_2_ increases, a gradual decrease in photosynthesis by RuBP regeneration or TPU restrictions is expected [32]. Afterwards, the rise of internal CO_2_ partial pressure (Ci) and the accumulation of specific metabolites can culminate in a series of negative feedback regulations, such as a decrease in stomatal conductance (g_S_) and stomatal density and the inhibition of photosynthesis-related proteins [33,34].

However, many studies over the decades have shown that the behavior of g_S_ on e[CO_2_] is contradictory and species-specific. Ainsworth and Rogers (2007) reported in a meta-analysis study from FACE (free-air concentration enrichment) experiments that e[CO_2_] decreased g_S_ in all plant groups, but to less extent in trees compared to grasses and herbaceous crops [27]. In a previous meta-analysis work, specific to woody plants, Medlyn et al. (2001) found no reduction in g_S_ after a short-term e[CO_2_] exposure (less than 1 year), while under long-term exposure (more than 1 year), g_S_ decreased 23% [35]. Additionally, they reported that conifers are less sensitive to e[CO_2_] than deciduous and evergreen broadleaf species. Notably, some studies reported that particular coniferous trees (*Pinus taeda* and *Pinus densiflora*) have guard cells insensitive to e[CO_2_] [36,37]. This response corroborates the finding that g_S_-[CO_2_] sensitivity increased as tree species evolved (gymnosperms < deciduous angiosperms < evergreen angiosperms) due to atmospheric CO_2_ level changes over the years [20]. Purcell et al. (2018), using in situ measurements of 51 woody plants, demonstrate that g_S_ can be increased in response to e[CO_2_] in specific weather conditions (high temperature and low humidity) depending on water availability [38]. Therefore, the downregulation of g_S_ under e[CO_2_] in some studies might be more related to methodological artefacts involving differences in the climatic and/or measurements conditions, water status and/or nutrients and signal-to-noise ratio of g_S_ [39] than an increase of Ci itself in the mesophyll and guard cells.

Stomatal size and density are other traits that could interfere with photosynthetic activity, but there is no clear evidence of this in woody plants in response to CO_2_ enrichment [27,40]. In *Coffea* exposed to long-term e[CO_2_], stomatal density and size showed dichotomous behavior, decreasing and increasing, respectively, without significant negative impacts on g_S_ and CO_2_ assimilation [40]. An increase in leaf area index (LAI) is observed in growing season trees, which can offset any downregulation of g_S_ on CO_2_ assimilation [41]. Nevertheless, after archive maturity, LAI tends to decrease in the upper layer of the canopy, reducing the shade effect and favoring light capture and photosynthesis in the whole shoot [25,41]. The coordination between g_S_ and the photosynthetic rate was described in many plant species grown in several environmental conditions [42]. It is clear that the stomatal aperture should maximize CO_2_ uptake while minimizing water loss to increase photosynthesis and water use efficiency. However, the connection between g_S_ and CO_2_ assimilation is diverse in woody plants, and while stomatal conductance does not seem to acclimate, photosynthesis does under e[CO_2_] [35].

### 2.2. Effect on Source–Sink Relationship and Nitrogen Metabolism

Photosynthetic downregulation after a long period of e[CO_2_] exposure has been termed CO_2_ acclimation, which was demonstrated in some species [43,44]. This response might be associated with the negative regulation of photosynthesis-related proteins, especially Rubisco, as in many cases reductions in V_cmax_ (*in vivo* Rubisco maximum carboxylation rate) are observed [44]. One hypothesis for this photosynthesis acclimation is the unbalance between source and sink activity. The accumulation of carbohydrates in source leaves (photosynthetic active) by the low utilization in sink tissues (non-photosynthetic) downregulates photosynthesis owing to negative feedback, decreasing the amount and activity of photosynthetic proteins [28,45,46]. The stimulation of photosynthesis by higher sink activity (root biomass) was observed in cassava (*Manihot esculenta* Crantz.) after e[CO_2_] acclimation [47]. Indeed, the export of photosynthates might increase to sink tissues by e[CO_2_] as plant growth increase and the expression/activity of sugar metabolism-related enzymes change [47,48,49].

A second hypothesis is related to nutrient dilution/acquisition, mainly nitrogen (N) and phosphorus (P), as a consequence of the rapid growth and/or allocation in sink tissues [50,51,52]. Rubisco is the most abundant protein in plants; then, a strong N-sink and reductions in its amount can be a common symptom of N deficiency and/or remobilization to other pathways, directly decreasing CO_2_ assimilation. Feng et al. (2015) reported a negative correlation between e[CO_2_] and plant N concentration in different ecosystems, including croplands, grasslands and forests. In this study, N limitation was more associated with the negative effects of e[CO_2_] on plant N uptake by unknown mechanisms rather than growth dilution [53]. The C/N unbalanced ratio seems to be related to the inhibition of photorespiration in plants exposed to e[CO_2_], as it was observed in a metabolic model that the levels of glycine and serine are correlated with *de novo* N assimilation [54]. However, the mechanisms underlying how e[CO_2_] limits N acquisition and if it is a cause or a consequence of photosynthesis acclimation are still open questions.

Actually, photosynthesis acclimation after e[CO_2_] exposure can be found in many plant species, especially in non-woody crops [46,55]. Nevertheless, it is a rare event in woody plants growing in normal conditions with no limitations of root growth space, water and nutrients [20,41]. For instance, tree species from northeast Asia grown under FACE conditions in infertile and immature volcanic ash (VA) and fertile brown forest (BF) soils for 2 years had different photosynthetic responses [56]. The photosynthesis of *Betula platyphylla* was downregulated in both soils, while in *Betula maximowicziana*, this response happened only in VA soil, probably due to the reduced N and Rubisco content. In contrast, a negative regulation in *Alnus hirsute* was observed only in BF soil, which might be related to higher amounts of starch in the leaves [56]. In many other studies, the photosynthetic rate in woody plants does not seem to acclimate, neither presenting any down-regulation nor even up-regulation [22,24,57,58].

High CO_2_ enhanced overall photosynthesis and Rubisco-specific activity, even with decreases in Rubisco content and photochemical parameters in *Betula pendula* [59,60]. A study with aspen tree (*Populus tremuloides*) exposed to e[CO_2_] revealed an increase in photosynthesis, despite the downregulation of many transcripts involved with chloroplast biosynthesis and function, including the photosynthetic protein genes Rubisco and Rubisco activase, proteins from photosystem I and II, light harvest complex and chlorophyll biosynthesis [61,62]. This transcriptional level pattern might be related to negative feedback triggered by the accumulation of sugars in leaves [62].

The carbohydrate level in source tissues is controlled by sink pathways (growth rate, respiration and storage/compartmentalization in certain organs), which tend to increase in high CO_2_-acclimated trees to keep a metabolic balance between synthesis and consumption. Respiration rate also responds in different ways according to the species and environmental conditions, and there is no immediate effect of high CO_2_ on mitochondrial respiration rate [63,64]. This process has been strongly increased in *Eucalyptus saligna*, while in *Coffea* spp. it has not been affected under e[CO_2_] [40,65]. Additionally, it has been demonstrated that leaf mitochondrial respiration could be unaffected by e[CO_2_], especially at night-time under mild temperatures, but the overall rate might be higher considering the whole plant leaf area, which usually increases in those conditions [65,66]. Some studies noticed a stimulation of gene expression and metabolite alterations of the TCA (tricarboxylic acid) cycle, such as citric, succinic, fumaric and malate acid in e[CO_2_]-enriched leaves [67,68].

In fact, the respiratory process in leaves should be slightly affected considering the photosynthetic stimulation in elevated CO_2_-exposed plants, as these are opposite reactions and strictly regulated. It is hypothesized that leaf respiration might be stimulated due to the higher concentration of non-structural carbohydrates (respiratory substrate), whereas the N dilution induced by elevated CO_2_ might reduce protein turnover and the demand for respiratory energy [65,69]. However, what has been reported to e[CO_2_] acclimated forest is an increase in fine root growth and rhizosphere respiration, indicating the export of photosynthates and utilization of heterotrophic respiration [7,70].

Therefore, the increase of belowground biomass and soil respiration might be a reason to support higher CO_2_ assimilation in trees fertilized with e[CO_2_] under non-stressful conditions, allowing a narrow connection between photosynthesis and ecosystem respiration [7,70]. The up-regulation expression of respiratory genes triggered by e[CO_2_] has been reported in leaves of non-woody plants [69,71]. However, the effect of e[CO_2_] focused on the respiration process at the transcriptional level of tree species is poorly investigated, particularly in roots.

Certainly, the behavior of source activity will depend on sink strength and how the whole plant metabolism will be adjusted to grow with the available resources. It is clear in the literature that woody plants will cope with e[CO_2_] by remodeling their metabolism to decrease the expression of key photosynthetic proteins, adjust nutrient distribution between the tissues and increase CO_2_ assimilatory capacity.

### 2.3. Growth and Developmental Stage-Dependent Regulation

The pattern of growth and establishment of woody plants exposed to e[CO_2_] will also depend on a wide range of factors—ultimately on species, genotype, developmental stage and environment. Studies linking physiological and molecular data are better documented in *Populus genera*, which is considered a model tree genus, than in other species [61,62,72,73,74,75,76]. In general, these works report photosynthesis and above-ground biomass stimulation with increases in stem wood density in high CO_2_-fertilized *Populus*, except for the triploid white poplar, which had lower photosynthesis but a higher stem diameter. The enhancement of shoot growth by e[CO_2_] was also observed in *Pinus radiata* [77], *Pinus sylvestris* [78], *Acacia karroo* and *Acacia nilotica* [22] and *Aulonemia aristulata* [58]. Along with this physiological response, the expression of genes from different categories in leaves and stems was strongly changed in *P. deltoides* [74]. In this work, stems showed more CO_2_-responsive transcripts than leaves (2.5-fold upregulated and 6.5-fold down-regulated in stems compared to leaves), and most of them were related to metabolism. The main enhanced genes in leaves were those involved with storage proteins and wall expansion, whereas in stems, they were more related to lignin biosynthesis (enzymes responsible for lignin formation and polymerisation and ethylene response factors), cell wall formation and cell growth, corroborating the growth stimulation response.

The effects of e[CO_2_] in plant metabolism also rely on the developmental stage of each species [61,79,80]. The gene expression pattern was dependent on leaf age in elevated CO_2_-acclimated *Populus* [61]. While in young leaves, the most differentially expressed genes were upregulated, in semimature leaves, were downregulated under e[CO_2_]. This environmental condition upregulated 16-photosynthetic gene transcripts, including Rubisco small subunit, in young leaves compared to older leaves [61]. Certainly, the predicted elevated CO_2_ atmosphere concentration will change the carbon (C) status in plants. Some studies have suggested that the leaf development pattern is likely to be accelerated by e[CO_2_], as gene expression related to photosynthesis, cell-wall loss and synthesis (xyloglucan endotransglycosylase) and calcium-signaling (CPK2) were up-regulated in young leaves [61,81]. These findings support the hypothesis of growth promotion and sucrose-cleaving and synthesizing enzyme activity modifications by CPKs under e[CO_2_] [82,83]. Moreover, in mature leaves, the C flux can be redirected to the glycolysis pathway once the transcripts for adenylate kinase are upregulated [61].

A commonly raised question would be whether leaf senescence is intensified in woody CO_2_-fertilized plants, once growth and LAI are generally increased. Different groups demonstrated that senescence-related gene (β-amylase and metallothioneins) expression and proteins were lower and, consequently, leaf longevity was augmented in e[CO_2_]-acclimated plants [61,62,73,80]. Moreover, a transcriptome study in *Populus* demonstrated that delayed senescence was correlated with the up-regulation of glycolysis and secondary metabolism genes, including anthocyanin biosynthesis-related genes (leucoanthocyanidin dioxygenase–LDOX, and dihydroflavonol reductase–DFR) [75].

Therefore, despite the limited studies, the findings obtained so far with woody species suggest that e[CO_2_] will increase plant growth through modifications in C allocation to different pathways (secondary metabolism) and organs (stem) and will also affect the ecosystem of soil microorganisms by enhancing respiratory substrates (fine roots turnover) [61,73,74,75]. Besides this, lignin accumulation in stems could contribute to pathogen resistance but may negatively impact the wood quality for both timber and paper production [74]. Hence, CO_2_-fertilized trees may not act directly to C sequestration, and the metabolic pathway changes may not be desired by agriculture. To better understand how e[CO_2_] will affect woody plant metabolism, more studies are needed, especially under the projected climate change scenarios.

## 3. Integrative Responses of Woody Plants under Elevated CO_2_ and Other Abiotic Stresses

Taking the main impact of e[CO_2_] on woody plant metabolism, photosynthesis and growth will tend to increase, as mentioned before. However, in addition to the accumulation of atmospheric [CO_2_], warm days and nights and an increased frequency of heat waves and drought, which contribute to land salinization and decrease in soil fertility, are expected worldwide [6,84]. These abiotic stresses, in addition to O_3_ tropospheric accumulation (secondary pollutant), are considered the primary causes of crop loss around the globe, reducing average yields and quality [18,85,86]. To grow and develop in such adverse conditions, plant metabolism must be completely rearranged, including changes at morphological, physiological, biochemical and molecular levels. The main findings obtained so far with woody plants exposed to the predicted climate changes are detailed in this work, aiming to answer the following questions: Will e[CO_2_] mitigate the negative effects of abiotic stresses? How will woody plants face multiple environmental adverse conditions?

### 3.1. Effects of Elevated CO_2_ and Heat Stress

The exponential accumulation of CO_2_ concentration in the atmosphere is considered the main driver of global warming. Since the pre-industrial era, along with the 129 ppm CO_2_ increase, the temperature has increased by 0.85 °C. According to the Intergovernmental Panel on Climate Change [6,87], it is expected that atmospheric CO_2_ levels will increase from the current 412 ppm to 936 ppm along with warmer weather, with temperatures rising between 0.3–1.7 °C in a mild scenario or even 2.6–4.8 °C in a severe scenario. Plant growth and development are directly dependent on air temperature, which determines the distribution of species around the planet [88]. However, supra-optimal temperatures lead to heat stress, which restricts plant growth and productivity by affecting water relations, membrane fluidity and stability (with negative impacts on the chloroplast and mitochondria reactions) and the whole metabolism [89,90]. In order to control water status under high temperatures, plants often reduce stomatal aperture, which further decreases CO_2_ flux and photosynthesis [91]. Moreover, in that condition, gas solubility increases, affecting the availability of O_2_/CO_2_ next to Rubisco active sites, which stimulate photorespiration and respiration, modifying plant energetic metabolism [27].

Therefore, it can be hypothesized that the negative impact of high temperature could be mitigated or plant growth could even be improved under the simultaneous increase of CO_2_, since e[CO_2_] increases photosynthesis and water use efficiency. In fact, many studies with different plant species reported that e[CO_2_] positively influenced photosynthesis and biomass yield under high-temperature conditions [15,92,93,94,95,96]. Despite the investigation of this, environmental studies involving trees are scarce and focused on physiological and biochemical approaches, although they clearly suggest that these species could take advantage of e[CO_2_] under higher temperatures if water and nutrient supplies and root growth space were sufficient. A study with two Eucalyptus species (*E. saligna* and *E. sideroxylon*) seedlings exposed to elevated CO_2_ (650 ppm) and temperature (ambient +4 °C) showed that each isolated condition stimulated photosynthesis and growth by increasing light-saturated photosynthesis, light and CO_2_-saturated photosynthesis and a maximal electron transport rate with no interaction between e[CO_2_] and high-temperature treatments [93,97]. Despite the lack of interaction, these results indicate that, in the future, plants such as eucalyptus could take advantage of e[CO_2_] under high temperatures by enhancing growth and potentially performing C sequestration if water and nutrients were sufficiently supplied.

It was demonstrated that e[CO_2_] ameliorated the negative impacts of high temperature in two *Coffea* species: *C. arabica* and *C. canephora* [96,98,99]. In these works, the combination of e[CO_2_] and high temperature (from 25 °C to 42 °C) improved photochemical efficiency, energy use and biochemical functioning in both species, particularly in the warmer condition, when compared to plants grown at ambient [CO_2_]. The better performance of coffee plants at higher temperatures and e[CO_2_] was related to the induction of protective molecules, such as antioxidants and molecular chaperones, which favored the maintenance of ROS at controlled levels and metabolism function [98]. Furthermore, Rodrigues et al. (2016) and Martins et al. (2016) reported that *C. arabica* and *C. canephora* are heat tolerant up to 37 °C, and irreversible effects of the extreme temperature threshold (42 °C) were strongly attenuated by e[CO_2_] [96,98]. The transcriptomic analysis in these *Coffea* genotypes exposed to the combination of heat and e[CO_2_] revealed a genotype-dependent gene expression [99].

This environmental condition resulted in differential expressed genes (DEGs) related to photosynthetic and biochemical processes. At the highest temperature (42 °C), the most responsive genes were down-regulated, especially in *C. arabica* rather than *C. canephora*. These DEGs were related to specialized metabolism, the lignin catabolic process in *C. arabica* (suggesting a decrease in lignin synthesis) and to molecular functions linked to calcium ion binding, oxidoreductase, sulfotransferase and xyloglucan:xyloglucosyl transferase activity in *C. canephora* (suggesting a decrease in cellulose synthesis). Moreover, the DEGs involved with photosynthetic and chlorophyll metabolic processes were up-regulated at 37 °C and down-regulated at 42 °C in *C. arabica* and down-regulated in both elevated temperatures in *C. canephora* compared to 25 °C [99] The down-regulated genes at 42 °C were related to photosynthesis, including Rubisco and Rubisco activase, and proteins involved with PSII assembly, stability and repair (PsbQ and PsbP). The authors reinforced the mitigating role of e[CO_2_] to extreme heat by keeping higher photosynthetic performance than plants under ambient CO_2_, especially in *C. canephora* plants [99]. Altogether, these results reveal that e[CO_2_] mitigates the effects of supra-optimal temperatures by keeping an integrated metabolism between photochemical efficiency, CO_2_ assimilation and specialized metabolism. These sink energy processes coupled with the stimulation of antioxidant pathways might be related to the maintenance of thylakoid membrane proteins and the maximum PSII quantum efficiency (Fv/Fm) under heat stress.

The effect of heat and e[CO_2_] on *C. arabica* bean quality was also investigated by Ramalho et al. (2018) [100]. It was observed that elevated temperature depreciates bean quality, but this effect was attenuated by the interaction with e[CO_2_], which kept bean properties closer to or even better than (higher acidity and more stable levels of O-caffeoylquinic acids) that obtained under control conditions. Therefore, woody plants might take advantage of e[CO_2_] to deal with mild high temperatures by enhancing photosynthetic machinery, cell wall composition and specialized metabolites that could be involved in stress signaling and defense.

### 3.2. Effects of Elevated CO_2_ and Drought

Climate change is completely altering the way we cultivate crops, as precipitation patterns are becoming scarce and drought events more frequent [6]. Furthermore, the expected higher temperatures, as mentioned above, will increase the leaf-to-air vapor pressure deficit (VPD), intensifying the dry effects on plants [101]. The increase in frequency and duration of drought has been associated with tree growth declines, mortality and regeneration reduction worldwide [102,103,104]. Stomatal closure, in response to abscisic acid (ABA) accumulation on leaves, is the initial response triggered by drought aiming to minimize water loss [105]. However, this response also contributes to decreasing CO_2_ assimilation, which may lead to insufficient carbohydrate supply for metabolism and cause plant death [106]. Therefore, it can also be hypothesized that the enrichment of CO_2_ ameliorates the negative effects of drought, taking into account the benefits of e[CO_2_] already mentioned. To test this hypothesis, some studies were performed employing physiological and biochemical approaches in field or greenhouse conditions.

However, it seems that some species will not benefit from the future predicted conditions, as demonstrated by *Populus deltoides* and *Liquidambar styraciflua* (sweetgum) [107,108]. In fact, the benefits caused by e[CO_2_] to decrease water loss by reducing stomatal conductance can be offset by the leaf area increase, which can be noticed in some woody plants [107]. Hence, Bobich et al. (2010) noticed that *P. deltoides* under e[CO_2_] in the field would be more susceptible to drought, as in that condition, these trees showed greater stomatal density and lower wood density than trees under ambient CO_2_ [107]. Both traits may facilitate xylem cavitation under severe drought. Later, it was demonstrated that e[CO_2_] promoted higher leaf senescence and abscission in temperate trees of sweetgum under severe summer drought when compared with ambient CO_2_ conditions [108]. In this study, the authors also showed that the canopy conductance and modelled photosynthesis were lower in stands exposed to e[CO_2_] than in ambient CO_2_. These responses could reduce latent heat loss and thereby elevate leaf temperature, which may decrease photosynthesis and increase respiration and further exacerbate the negative effects of drought [108].

It is worth considering that these first investigations were performed under natural field conditions in southern Arizona and eastern Tennessee (USA), in which the effect of e[CO_2_] was combined with multiple stresses (drought, high temperature and high deficit pressure vapor) characterizing severe drought. Moreover, these responses involve only two species with no well-watered stand to compare with, and some of the parameters were estimated and not properly measured. Therefore, more studies are needed to better understand how woody plants will cope with e[CO_2_] and drought in the near future.

Later, Bachofen et al. (2018) reported that pine seedlings (*Pinus sylvestris* and *Pinus nigra*) acclimated to high CO_2_ were not more susceptible to drought than plants grown under ambient CO_2_ [109]. These plant groups showed similar starch concentration, biomass production and mortality when subjected to water deficit in a semi-controlled condition. Another recent research work verified that drought-stressed *Pinus halepensis* (Aleppo pine) fertilized with e[CO_2_] had higher total biomass, net photosynthesis, water use efficiency and water potential, without changes in respiration rate, compared to ambient CO_2_-acclimated plants [110]. These results also concur with what has been reported on *Coffea arabica* plants under controlled or natural conditions [49,68,111,112,113,114]. In these studies, e[CO_2_] mitigated the negative effects of water deficit by increasing CO_2_ assimilation, photochemical efficiency, water use efficiency and soluble solutes content, decreasing photorespiration rate and oxidative pressure and keeping hydraulic conductance. The increase of water use efficiency is a common response triggered by e[CO_2_], as plants can keep lower stomatal conductance to uptake atmospheric CO_2_ without increasing leaf transpiration. Therefore, woody plant growth under drought is benefited by e[CO_2_], as already reported [68,110,111,113,114]. Moreover, the buffer effect of e[CO_2_] under drought conditions could be related to changes in specialized metabolites such as 5-O-caffeoylquinic acid (5-CQA) and caffeine [113].

Recently, studies have demonstrated that the buffering effect of e[CO_2_] in *Coffea* plants exposed to different levels of water deficit is genotype-dependent [114,115]. Elevated CO_2_ attenuated the negative drought impacts in *C. canephora* cv. Conilon and maintained the high tolerance to drought in *C. arabica* cv. Icatu plants [114]. While in Conilon plants, e[CO_2_] improved the photochemical efficiency by increasing the PSII efficiency and decreasing the thermal energy dissipation, this beneficial effect in Icatu plants was slight and more related to keeping the abundance of photosynthetic proteins and the cyclic electron flow at PSI under moderate or severe drought conditions [114]. The contrasting effect of e[CO_2_] in these *Coffea* genotypes exposed to different water deficit conditions was also noticed at the metabolic level [115]. While the abundance of the most primary metabolites (mainly amino acids) decreased in Conilon, they increased in Icatu exposed to the combination of e[CO_2_] and drought [115]. The metabolite profile was more affected by severe water deficit regardless of CO_2_ in both genotypes, except for Icatu plants under moderate water deficit and e[CO_2_], which were close to well-watered plants, suggesting a beneficial effect of CO_2_ enrichment in moderate drought conditions.

### 3.3. Effects of Elevated CO_2_ and Salinity

Land salinization is one of the major abiotic stresses that negatively impact plant growth and survival, especially in semiarid and arid regions [116]. This problem is increasing worldwide as a consequence of climate change, deforestation and irrigation practices [117,118,119]. Soil salinity is a serious threat to agriculture that already affects more than 6% of the total land area and 30% of the irrigated farmland around the globe [120,121]. Despite also being a problem for native plants and non-agriculture lands, the effects of salt stress are much less recognized and investigated in natural systems [122]. Plant physiology and development are adversely impacted by salinity through both osmotic and ionic stresses. In general, the accumulation of solutes in the soil limits root water uptake, decreasing photosynthesis, transpiration and plant growth [123]. Further, the high concentration of ions (i.e., Na^+^ and Cl^−^) in leaves can lead to an alteration of K^+^ homeostasis and disrupt plant biochemical processes [124]. This condition can alter cellular protein structure, reduce photosynthesis and increase reactive oxygen species (ROS) [124].

The effect of salinity in woody plants is poorly studied, and much less is known about the interaction between this abiotic stress and e[CO_2_] and how this species will cope with this condition in the future. Investigating the simultaneous supply of e[CO_2_] and salinity on two cultivars of *Olea europaea* L. (cv. Picual–salt tolerant and cv. Koroneiki–salt sensitive), it was observed that salt stress decreased the positive effects of e[CO_2_], especially in the sensitive cultivar [125]. Additionally, e[CO_2_] enhanced WUE under salt stress on both cultivars and decreased salt ion uptake in the sensitive cultivar. A short treatment with high CO_2_ also alleviated the toxic effects of salt stress, favoring a higher K^+^/Na^+^ ratio than ambient CO_2_ cashew (*Anacardium occidentale*) plants [126]. In that study, e[CO_2_] decreased the activity of glycolate oxidase and the concentration of hydrogen peroxide (H_2_O_2_), NH_4_^+^ and glyoxylate in salt-stressed plants, suggesting a higher photoprotection and lower photorespiration rate than plants acclimated to ambient CO_2_. These responses related to the buffer effects of e[CO_2_] under salt stress corroborate with what has been found in non-woody plants, which showed better control of water status, osmotic adjustment and antioxidant activity [127,128,129].

### 3.4. Effects of Elevated CO_2_ and Ozone

Besides the accumulation of atmospheric CO_2_ amount, the increase of tropospheric ozone concentration (O_3_) has also been noticed, contributing to the greenhouse effect and negatively impacting plant growth [18]. This gas is a secondary pollutant synthesized from the photochemical oxidation of NO_x_ in the presence of carbon monoxide (CO), methane (CH_4_) and non-methane hydrocarbons [130]. Tropospheric O_3_ can increase in the future, as the expected climate change scenarios (summer droughts and heatwaves conditions) favor its accumulation [86]. In general, high O_3_ induces oxidative stress in plants, which has contributed to large decreases in gross primary productivity in Europe (3–9%), China (10–18%) and the U.S. (5–10% for soybean and maize) in the last decades [131,132,133]. Therefore, considering the positive effects of e[CO_2_], some studies were developed aiming to investigate whether CO_2_ could ameliorate the stress induced by high O_3_ in the woody species aspen and birch.

In fact, high O_3_ caused foliar damage in *Populus tremuloides* due to reductions in chlorophyll, carotenoid, starch and Rubisco concentration and the transcription level of Rubisco small subunit [21]. The plants also showed higher activity of antioxidant enzymes (ascorbate peroxidase, catalase and glutathione reductase), phenylalanine ammonia-lyase (PAL) and 1-aminocyclopropane-1-carboxylic acid (ACC)-oxidase transcript levels. The combination with e[CO_2_] decreased the visible leaf injury and leaf growth but did not offset the main harmful effects of O_3_ as the activity of antioxidant enzymes and PAL decreased, which could exacerbate the stress by high O_3_ exposure in aspen [21]. The long acclimation of *P. tremuloides* to e[CO_2_] and high O_3_ revealed a different pattern of transcripts expression when analyzed either isolated or combined [72]. While e[CO_2_] up-regulated only small numbers of genes, high O_3_ increased the expression of many signaling and defense-related genes and decreased the expression of several photosynthetic and energy-related genes. Surprisingly, the combination of treatments (CO_2_ + O_3_) resulted in the differential expression of genes that were not up-regulated under the isolated conditions.

The studies involving birch (*Betula pendula* and *Betula papyrifera*) demonstrated that the effect of elevated O_3_ was variable and clone-dependent [59,60,80,134,135]. In those works, some sensitivity clones showed reductions in photosynthetic rate and advanced leaf senescence when exposed to high O_3_. The combination with e[CO_2_] had a synergistic effect and partially buffered the stress caused by O_3_ in the plants. This response was related to the higher carbon/nitrogen ratio, in which the greater quantity of carbohydrates may be used to detoxify and repair cell damage under elevated O_3_ and CO_2_ [80,134]. The buffer effect of e[CO_2_] in plants exposed to high O_3_ can also be involved with the decreased stomatal conductance triggered by the first gas, allowing a smaller amount of O_3_ flux into leaves and therefore less damage [21,136]. In addition to that, the lower O_3_ inside the leaves may less effectively induce the antioxidant system when combined with e[CO_2_] than under isolated high O_3_ [80].

The combination of elevated CO_2_ and O_3_ also increased the gene expression of proteins related to glycolysis, suggesting the requirement of energy and carbon skeletons for biosynthetic pathways and differential expression of genes related to steroid biosynthesis (increase of farnesyl-PP and phytyl-PP and decrease of pre-squalene and squalene) [80]. Indeed, the negative effects of O_3_ were smaller when combined with e[CO_2_], indicating that both gases induce a synergistic response in woody plants when applied simultaneously. Moreover, the studies with long-term exposure demonstrated that O_3_ sensitivity changes according to the growth stage and environmental conditions (season), where stressful conditions increase the O_3_ sensitivity [59,134]. The alleviation of O_3_ stress by e[CO_2_] can also be temporary, as demonstrated on *Betula pendula,* which showed a gene expression, phenology and physiology in August similar to plants exposed to e[CO_2_] alone, while in September, these traits were similar to plants exposed to high O_3_ alone [135].

### 3.5. Effects of Elevated CO_2_ and Multiple Stresses

Most of the studies mentioned above in this review considered the effect of e[CO_2_] combined with one abiotic stressor. The main responses associated or in common with specific conditions are highlighted in Figure 1. In general, the physiological responses are supported by biochemical and molecular data. Elevated CO_2_ decreases the negative effects of stress, such as photorespiration, oxidative stress, salt uptake and leaf senescence, and increases photosynthetic and water use efficiencies, plant biomass, the antioxidant system and the synthesis of carbohydrates, amino acids and specialized metabolites.

However, trees are constantly exposed to multiple stresses in the natural environment, and the effects on plant metabolism between e[CO_2_] and multiple stresses might be different. This condition was investigated in *Alnus hirsuta* and *Alnus maximowiczii* under e[CO_2_], phosphorus (P) availability and drought, which demonstrated the benefit of e[CO_2_] on N_2_ fixation by increasing nodule biomass under a high P supply [137]. This positive effect was absent under low P concentration, indicating that the e[CO_2_]-buffer effect is dependent on nutrient supplementation. This corroborates what has been found in *Eucalyptus tereticornis*, the growth of which was improved by e[CO_2_] under heat and sufficient soil P concentration [138]. The mitigating effect of e[CO_2_] in *Quercus rubra* and *Eucalyptus* seedlings exposed to high temperature and low water supply was reported by Bauweraerts et al. (2013) and Lewis et al. (2013), respectively [19,139]. It was observed that the extreme heat events differently affect growth and gas exchange when compared to constant high temperature, reinforcing the idea that the environmental fluctuations are important to consider to better understand plant performance in their natural condition [139]. Additionally, it was noticed that e[CO_2_] buffered the effect of heat treatments regardless of water availability, but this response is plant species and stress level-dependent [19,139].

Therefore, elevated e[CO_2_] buffers the negative effects of mild/moderate abiotic stress by improving the photosynthetic rate through Rubisco activity and increasing the concentration of primary metabolites. Possible negative feedback regulates stomatal aperture and increases water use efficiency. Moreover, there is evidence for alterations in other metabolic pathways, such as the enhancement of aquaporin genes, protective proteins (chaperones and antioxidative system), specialized metabolites, nutrient requirement, changes in the respiratory rate and a decrease of genes and proteins involved with leaf senescence. However, the mechanisms involved in these processes are poorly understood, especially the molecular bases. As the stress becomes severe, the beneficial role of e[CO_2_] is offset, probably as a consequence of stomatal closure and the severe cell damage caused by the stress (Figure 2).

## 4. Concluding Remarks

This review highlighted the main effects of elevated CO_2_ alone and combined with abiotic stresses in woody plants, focusing on the understanding of the extent to which e[CO_2_] should mitigate the predicted climate change scenarios and how molecular mechanisms should respond to these environments. In general, e[CO_2_] increases photosynthetic efficiency, with little or no photosynthetic acclimation, and biomass and buffers the negative effects of most abiotic stresses when they are not severe and no growth limitations are imposed, such as root growth and nutrient availability (Figure 1 and Figure 2).

The integration of modern omics (genomics, transcriptomics, proteomics, metabolomics and ionomics) along with gene editing offered a better understanding of the genetic and molecular basis of staple crops, allowing the production/selection of more resilient and productive cultivars [140]. However, these molecular approaches have been poorly applied to woody plants, especially involving the interaction with multiple abiotic stresses, making it unclear how the increase of atmospheric CO_2_ will affect tree species and forest ecosystems. Therefore, more studies that supply an integrative view of powerful tools (physiological, biochemical and molecular) and the combination of e[CO_2_] and abiotic stresses are needed to better understand the effects of these conditions on woody plant metabolism. Finally, this deep level of knowledge could be applied to select/produce more resilient and productive plants and also to create strategies to maintain agricultural and forest productivity and environmental sustainability.

## Figures and Tables

**Figure 1 plants-11-01880-f001:**
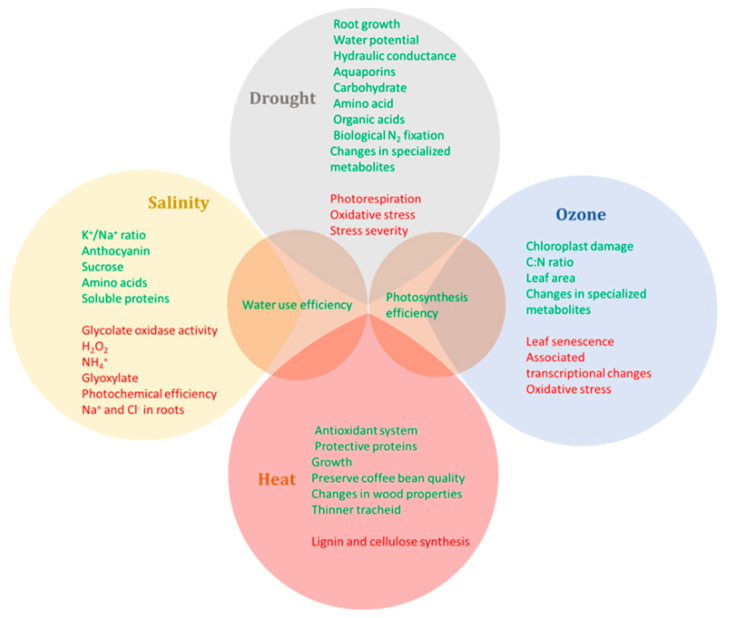
Venn diagram of the main effects caused by elevated CO_2_ in combination with abiotic stresses (heat, drought, salinity and ozone) in woody plants. Green effects are increased or stimulated, and red effects are decreased or inhibited.

**Figure 2 plants-11-01880-f002:**
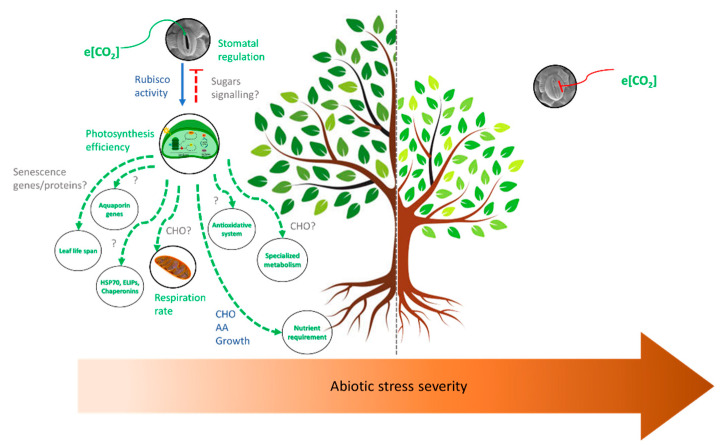
The main effects triggered by elevated CO_2_ according to abiotic stress severity. Elevated CO_2_ buffers the negative effects of mild/moderate abiotic stresses by inducing a range of molecular and metabolic changes. The positive response is eliminated under severe stress where the stomata is closed and photosynthesis and other processes are impaired. CHO: carbohydrate; AA: amino acids.

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
