# Peer review of "Physiological and Molecular Responses of Woody Plants Exposed to Future Atmospheric CO2 Levels under Abiotic Stresses"

_plants, 2022, doi:10.3390/plants11141880_

Round 1
Reviewer 1 Report
In reading through this review article, it was not evident what the underlying objective was: to bring together a broad collection of articles from previously published journals without a clear sense of direction of what the authors were trying to get to; or to provide a critique of the current literature with an emphasis to identify strengths and weaknesses and areas of research. In looking over all of the various sections included by the authors, it appears they have achieved the former – just a spectrum of loosely joined topics of the author's choosing without any critique or advancement of the literature. There has been a broad range of articles on "plant responses to atmospheric CO2 levels due to climate change" published in the past several years. The authors have included some of these. The question is “ what is new and unique in this review that cannot be found elsewhere”. I would argue very little is novel. Therefore, I conclude that there is not sufficient originality in this paper to warrant publication.
Author Response
Dear reviewer
Thank you for your feedback. We made changes in order to make this review more clear and assertive.
Reviewer 2 Report
The review entitled; Physiological and molecular responses of woody plants exposed to the future atmospheric CO2 levels under abiotic stresses describes about the role atmospheric CO2 and different abiotic stresses to affect performance of woody plants. The review is written well where, approaches of morphology and of molecular physiology were used to explain the plant output and merits publication. However, I will suggest following changes before its acceptance.
Abstract
Please indicate a clear objective of current review
Introduction
Please indicate a clear hypothesis “how the objective” was met.
Section 2.2. Effect on source-sink relationship and nitrogen metabolism, please include some information about transport of photosynthates (carbohydrates) from source to sink. Authors can take help from following manuscripts:
Shokat S, Novák O, Široká J, Singh S, Gill KS, Roitsch T, Großkinsky DK, Liu F. (2021). Elevated CO2 modulates the effect of heat stress responses in Triticum aestivum by differential expression of isoflavone reductase-like (IRL) gene. Journal Experimental Botany. 72: 7594–7609.
Ulfat A, Shokat, S, Li X, Fang L, Großkinsky DK, Majid SA, Roitsch T, Liu F. (2021). Elevated carbon dioxide alleviates the negative impact of drought on wheat by modulating plant metabolism and physiology. Agricultural Water Management. 250: 106804.
Shokat S, Großkinsky DK, Roitsch T, Liu F. (2020). Activities of leaf and spike carbohydrate-metabolic and antioxidant enzymes are linked with yield performance in three spring wheat genotypes grown under well-watered and drought conditions. BMC Plant Biology. 20.400.
Li X, Ulfat A, Shokat S, Liu S, Zhu X, Liu F. (2019). Responses of carbohydrate metabolism enzymes in leaf and spike to CO2 elevation and nitrogen fertilization and their relations to grain yield in wheat. Environmental and Experimental Botany 164:149-156.
Section 3.1. Effects of elevated CO2 and heat stress, please include the role of chlorophyll fluorescence (fv/fm). Again, you can take help from above-mentioned articles.
Section 3.2. Effects of elevated CO2 and drought, please include some information (one paragraph) about water use efficiency.
Conclusion should be concise with clear take home message.
Author Response
First of all, thank you for the constructive feedback.
Replying your suggestions:
Abstract
Please indicate a clear objective of current review
Answer: We have included a clear objective in the Abstract.
Introduction
Please indicate a clear hypothesis “how the objective” was met.
Answer: We have included a clear hypothesis at the end of Introduction.
Section 2.2. Effect on source-sink relationship and nitrogen metabolism, please include some information about transport of photosynthates (carbohydrates) from source to sink. Authors can take help from following manuscripts:
(...)
Answer: We have included this information and the references.
Section 3.1. Effects of elevated CO2 and heat stress, please include the role of chlorophyll fluorescence (fv/fm). Again, you can take help from above-mentioned articles.
Answer: This information was referred as improvements in photochemical efficiency and increases of expression of photosynthetic and chlorophyll metabolic related-genes, as reported by references 91, 93 and 94. We have added the role of this response to plant growth.
We appreciated the above-mentioned papers, but as our review is focused on woody plants and this information is in the already cited references we decided to not include them.
Section 3.2. Effects of elevated CO2 and drought, please include some information (one paragraph) about water use efficiency.
Answer: We have included this information.
Conclusion should be concise with clear take home message.
Answer: We have rearranged conclusions.
Reviewer 3 Report
Dear Authors
This review highlighted the main effects of elevated [CO2] alone and combined with abiotic stresses in woody plants, focusing on the understanding of what extension e[CO2] should mitigate the predicted climate change scenarios and how molecular mechanisms should respond to these environments. Overall, e[CO2] stimulates photosynthesis and growth and attenuates mild to moderate abiotic stress in woody plants if root growth and nutrients are not limited.
Authors should supplement their work with additional more recent manuscripts 2021 and 2022; they should also highlight their point of view more at the end of each chapter.
Line 14: “represent the bottleneck to plant development and production”. Please rewrite.
Line 41: [6-9] Hartmann, H., Bastos, A., Das, A. J., Esquivel-Muelbert, A., Hammond, W. M., Martínez-Vilalta, J., ... & Allen, C. D. (2022). Climate change risks to global forest health: emergence of unexpected events of elevated tree mortality worldwide. Annual Review of Plant Biology, 73, 673-702.
Lines 43-44: Add the following sentence: “In this scenario, plant growth, phenologiacal fases, and development will be frequently challenged by climatic fluctuations, leading food security or food quality at risk in an increasing world population [10-12]”
[11] Cataldo, E. C., Salvi, L. S., Paoli, F. P., Fucile, M. F., Masciandaro, G. M., Manzi, D. M., ... & Mattii, G. B. M. (2021). Effects of natural clinoptilolite on physiology, water stress, sugar, and anthocyanin content in Sanforte (Vitis vinifera L.) young vineyard. The Journal of Agricultural Science, 159(7-8), 488-499.
[12] Cataldo, E., Fucile, M., & Mattii, G. B. (2022). Effects of Kaolin and Shading Net on the Ecophysiology and Berry Composition of Sauvignon Blanc Grapevines. Agriculture, 12(4), 491.
Line 134 croplands, grasslands, and forests.
Line 249 O3
Author Response
Dear referee,
Thank you for a positive feedback.
Here are our reply to specific questions:
Authors should supplement their work with additional more recent manuscripts 2021 and 2022; they should also highlight their point of view more at the end of each chapter.
Answer: We have incorporated this suggestion.
Line 14: “represent the bottleneck to plant development and production”. Please rewrite.
Answer: We have changed it.
Line 41: [6-9] Hartmann, H., Bastos, A., Das, A. J., Esquivel-Muelbert, A., Hammond, W. M., Martínez-Vilalta, J., ... & Allen, C. D. (2022). Climate change risks to global forest health: emergence of unexpected events of elevated tree mortality worldwide. Annual Review of Plant Biology, 73, 673-702.
Answer: We have included it.
Lines 43-44: Add the following sentence: “In this scenario, plant growth, phenologiacal fases, and development will be frequently challenged by climatic fluctuations, leading food security or food quality at risk in an increasing world population [10-12]”
[11] Cataldo, E. C., Salvi, L. S., Paoli, F. P., Fucile, M. F., Masciandaro, G. M., Manzi, D. M., ... & Mattii, G. B. M. (2021). Effects of natural clinoptilolite on physiology, water stress, sugar, and anthocyanin content in Sanforte (Vitis vinifera L.) young vineyard. The Journal of Agricultural Science, 159(7-8), 488-499.
[12] Cataldo, E., Fucile, M., & Mattii, G. B. (2022). Effects of Kaolin and Shading Net on the Ecophysiology and Berry Composition of Sauvignon Blanc Grapevines. Agriculture, 12(4), 491.
Answer: We have done the suggested changes.
Line 134 croplands, grasslands, and forests. ?
Line 249 O3
Answer: We corrected it.
Reviewer 4 Report
I consider that the review is useful and contributes to the development of knowledge through the synthesis.
However, I appreciate that the paper is difficult to follow and I recommend the improvement.
I recommend including at least 2 summary tables, which reduce the text and simplify the comparative analysis. For example, chapter 3.1, lines 83-100, chapter 3.1, lines 273-300. These are just suggestions, the authors can decide if other chapters are more suitable for synthesis in tables.
Author Response
Dear referee,
Thank you for the feedback. We have simplified the table and moved to the supplementary material, all in order to make the review easier to follow.
Round 2
Reviewer 1 Report
I think that the current version of the manuscript can be published in Plants